# Associations between Home Foreclosure and Health Outcomes in a Spanish City

**DOI:** 10.3390/ijerph16060981

**Published:** 2019-03-19

**Authors:** Mariola Bernal-Solano, Julia Bolívar-Muñoz, Inmaculada Mateo-Rodríguez, Humbelina Robles-Ortega, Maria del Carmen Fernández-Santaella, José Luís Mata-Martín, Jaime Vila-Castellar, Antonio Daponte-Codina

**Affiliations:** 1Andalusian School of Public Health, 18080 Granada, Spain; mariola.bernal.solano@gmail.com (M.B.-S.); adapontec@gmail.com (J.B.-M.); inmaculada.mateo.easp@juntadeandalucia.es (I.M.-R.); 2CIBER Epidemiology and Public Health (CIBERESP), 28029 Madrid, Spain; 3Andalusian Observatory on Environment and Health (OSMAN), 18080 Granada, Spain; 4Faculty of Psychology, University of Granada, 18071 Granada, Spain; hrobles@ugr.es (H.R.-O.); mcfersan@ugr.es (M.d.C.F.-S.); matamar@ugr.es (J.L.M.-M.); jvila@ugr.es (J.V.-C.)

**Keywords:** foreclosures, economic crisis, housing, social determinants of health, Spain

## Abstract

The financial crisis has caused an exponential increase of home foreclosures in Spain. Recent studies have shown the effects that foreclosures have on mental and physical health. This study explores these effects on a sample of adults in the city of Granada (Spain), in terms of socio-demographic, socio-economic and process characteristics. A cross-sectional survey was administered to obtain information on self-perceived changes in several indicators of physical and mental health, consumption of medications, health-related behaviors and use of health services. A total of 205 persons, going through a foreclosure process, participated in the study. 85.7% of the sample reported an increase of episodes of anxiety, depression, and stress; 82.6% sleep disturbances; 42.8% worsening of previous chronic conditions, and 40.8% an increase in consumption of medication. Women, married persons and persons already in the legal stage of the foreclosure process reported higher probability of worsening health according to several indicators, in comparison with men, not married, and individuals in the initial stages of the foreclosure process. The results of this study reveal a general deterioration of health associated with the foreclosure process. These results may help to identify factors to prevent poor health among populations going through a foreclosure process.

## 1. Introduction

The global financial crisis, which started in 2008, was closely linked to the housing market and has had a particularly strong impact in Spain. This impact was not homogeneous, since it affected the Spanish Mediterranean regions, such as Andalusia, to a much greater extent. This greater impact was due to the enormous role that housing construction has in the economy and employment in these regions, which are closely associated with tourist activities [1].

For example, in 2007, in Andalusia (which represents 18% of the Spanish population), some 125,000 new homes were completed. In comparison, the median home construction for all Spanish regions was 25,000 new homes [2]. 

The increase in unemployment from 13.8% in 2008 to 22.4% in 2015, the rise in the number of households with no income from 2.12% in 2007 to 3.5% in 2012, and the progressive worsening of child poverty according to United Nations Children’s Fund (UNICEF) [3,4] are factors which have led to the emergence of vulnerable situations which affect different sectors of the population. In 2014, 22.2% of the population in Spain was living below the poverty threshold, one million more than in 2008. In Southern Spain, Andalusia is one of the most economically disadvantaged regions in the country. The unemployment rate in Andalusia in 2015 reached 31.7% [3], and 42.3% of residents—51.1% of whom are children—were at risk of poverty and social exclusion [5]. 

Loss of housing also increased dramatically starting in 2008. This could be caused by the loss of property due to non-payment of mortgage, or the lack of use due to unpaid rents. Mortgage default losses caused, proportionally, the majority of the losses in the harshest years of the economic crisis (2008–2013). Since 2015, these proportions have been reversed. In fact, from 2008 to 2015, evidence shows that legal actions in Spanish courts related to housing were “dispossessions” of home ownership, rather than the withdrawal of use (loss of use), which is mainly due to non-payment of rentals [6]. 

Likewise, there is evidence indicating that foreclosures were not evenly distributed among the neighborhoods of the Spanish cities, but rather, they were clustered mainly in deprived neighborhoods. In addition, foreclosures have affected homes of smaller sizes and prices more frequently [7].

In countries where the economic crisis did not reach the intensity experienced by the countries of southern Europe, and where its effects on the lives of the population were less significant, the evictions still affected the already disadvantaged population. Evictions were concentrated in low-income and rental populations. They were an additional element to other living conditions of these vulnerable populations, which will affect them for a long time. Among the causes of evictions could be specific public policies applied in those contexts [8,9,10].

Economic austerity policies have worsened this situation, not only playing a part in the spread of poverty to the middle classes, but also intensifying it and creating a chronic problem [11]. This has been the backdrop for mass evictions and foreclosures, giving rise to an unprecedented housing crisis in a country where housing is often considered, and treated by governments and other social and economic institutions, as a speculation asset rather than a primary necessity. Spain has one of the highest rates of unoccupied dwellings in the European Union (EU) (13.7%) [12], and one of the lowest percentages of social housing for rent—less than 2% compared to 17% in a neighboring country like France, or 34% in Holland [13]. Meanwhile 646,681 foreclosures took place between 2008 and 2015 according to official data in Spain [14]. This figure is disproportionately high in comparison with figures recorded in other European countries such as England, Holland, or Sweden [15].

In Spain, the foreclosure process begins when homeowners have difficulties in paying the mortgage. This is the first phase of the process. During this phase, homeowners will receive letters or telephone communications from the bank claiming the amounts due, and in many cases, threatening foreclosure. After a variable time, between three and six months, according to some sources, if the mortgage payments were not made, then the bank will file a lawsuit in the courts, claiming due amounts. Then, what we call the “legal process” phase of the foreclosure begins. In this phase, negotiation is attempted between the bank and homeowners. If no solutions are found, the lawsuit proceeds and the court sentences the loss of home ownership. The process may be delayed for various reasons. In many cases, according to Spanish legislation, the obligation to pay the debt is maintained, even after the loss of property. Thus, in Spain, the process of foreclosure is complex and usually takes several years [16].

Therefore, there are two major phases. The first consists of the owners not complying with the mortgage payments regularly (problems in the payment of the mortgage). The second phase is when the process is formalized in the court, until its final resolution (legal phase). Although there are no precise statistics, some data indicate that the process can last on average from 2 to 3 years, or even more time [12,17].

Several studies have analyzed the impact of the current foreclosure crisis on health. Most of them have been carried out in the USA or in other EU countries and have focused on the impact on mental health [18,19,20,21]. Other studies have analyzed different health indicators, highlighting the worsening of perceived health [22] or smoking [23]. The studies carried out in Spain have shown a worsening of the general health among people who live through an eviction or foreclosure process [24,25,26,27,28] and, particularly, a worsening of their mental health as the process evolves [28,29].

The evidence generated so far is, however, insufficient, especially considering the specific characteristics of the foreclosure process in Spain, where housing has been promoted as a key element for economic growth, generating some anomalies that differentiate it from USA or other EU countries. These include: (1) the existence of a foreclosure legislation that allows banks to demand the payment of the mortgage debt in full, plus interests, commissions, and any other expenses, making the debt almost unpayable. Recently, this legislation has been considered abusive and even illegal by the Court of Justice of the European Union. (2) Banks can use an extrajudicial procedure to speed up the process, with fewer guarantees for the people affected; and (3) the foreclosure does not cancel the debt, which becomes permanent, even if the foreclosure occurs [16]. During this financial crisis, the core of the Spanish political system was aligned in favor of defending the interests of banks, which has made it much more difficult to find solutions to avoid foreclosures and the subsequent suffering of the affected population [30].

The present article seeks to take a step towards the explanation of this possible deprivation of health of the affected population. For this, we analyze and describe changes in health indicators of persons affected by the foreclosure process, and the socio-demographic, socio-economic and process characteristics associated with these changes. To this end, self-perceived changes in various indicators of mental and physical health, consumption of medication, health-related habits and use of health services are analyzed. 

## 2. Materials and Methods 

### 2.1. Context and Sample 

A cross-sectional survey was administered to a sample of persons aged 18 and older, affected by the process of losing their home. The survey collected information on self-perceived changes in several indicators of physical and mental health, as well as consumption of medications, health-related behaviors, and the use of health services. 

Participants were recruited among people attending the weekly meetings of the platform “Stop-Desahucios” (Stop-Evictions) of Granada. “Stop-Desahucios” is the Spanish platform for the support of people affected by the foreclosure or eviction process created in 2009. It provides help mainly through legal advice, organization of concentrations and protests, and negotiation with banks. During the study period, team members informed attendees of meetings about the objectives of this study and requested their participation. The field work was carried out in 2013 and 2014. This study was approved by the Ethics Committee for Research of Granada, in Spain.

### 2.2. Sources of Information

To expand on the questionnaire, in-depth interviews were previously conducted with some people in the process of foreclosure, as well as leaders of the Stop-Evictions platform, health and social workers, psychologists, and lawyers who provided advice to people in this situation. The questions related to health were selected from several population health surveys used in Spain. Finally, some specific scales of mental health were used. The questionnaire was piloted with a small sample of individuals [26].

Participants were asked if they noticed significant changes attributable to the foreclosure process. Specifically, they were asked: “have you observed, since the beginning of the process, significant changes attributable to the process of foreclosure?”. These changes referred to the state of health, consumption of medications, use of health services, and lifestyle. Likewise, questions about the foreclosure process, its stages and characteristics, as well as the triggering factors, were also included. 

We present here the results describing the changes in the health status of the study participants, attributable to the foreclosure process.

### 2.3. Analysis

For the analysis, responses to the questionnaire were recoded in two categories, depending on the specific condition. Dependent variables were grouped into four blocks:Health indicators: sleep patterns; chronic diseases; episodes of anxiety, depression and/or stress.Habits related to health: smoking; consumption of illegal substances; consumption of alcohol; consumption of vegetables; and physical activity.Medication: consumption of psychotropic medication; and consumption of other medication.Use of health services: visits to primary health care services; mental health appointments; and visits to emergency rooms.

Independent variables were grouped into three blocks:

(1) Socio-demographics: sex; age; place of birth; marital status; living with partner; number of persons living in the household; and employment situation.

(2) Socio-economics: socio-professional class [31]; educational level; current income; difficulties in making ends meet; and reduced income.

(3) Related to the process: the main reason the process was initiated; stage of process; and when the process began.

To characterize the sample, firstly, a descriptive analysis was made of each of the independent variables (socio-demographic, socio-economic, and process) for the total sample, and samples differentiated by sex. Secondly, proportions were estimated and compared for each of the health indicators for the total sample and for men and women separately, using contingency tables and the Chi-square test. 

Then, the odds ratio (OR) and 95% confidence intervals (CI) were estimated using logistic regression in order to study associations between each health indicator and each independent variable. Finally, we used penalized multivariate logistic regression to estimate adjusted OR and 95% CI. SPSS 25 software (IBM Corp., Armonk, NY, USA) was used for the analysis.

## 3. Results 

A total of 205 respondents affected by a foreclosure process took part in the study, of whom 59.5% (122) were women and 40.5% (83) were men. Most of them were aged between 36 and 50. A large majority was married or living with a partner (88.4%) and in households with 3–5 members (67.5%). Although the predominant profile was composed of persons with primary level education or lower (60.6%), unemployed (63.1%), or with monthly family income of 500 € or less (43.8%), also suffering foreclosure were persons with university education (9.9%), currently in paid work (26.2%), and with monthly incomes from 1000 to 3000 € (18.2%). Economic difficulties in getting through to the end of the month were reported by practically all the persons in the sample (97.9%), and the most frequent reason for being in a foreclosure process was job loss (58.1%) (Table 1).

Regarding the stage of the process, slightly more than half of the sample reported being in the initial stages, i.e., finding it very hard to meet payments or already behind on payments (53.9%). Although most of the cases were recent (67.8%), almost one-third had already been going through the process for more than three years (32.2%).

Regarding differences by sex, men were younger, 25 to 35 years old (34.9% vs. 23.8%), while there were almost three times as many women as men with university education (13.3% vs. 4.8%), in employment (33.3% vs. 16.0%), working in unskilled manual jobs (37.8% vs. 20.0%) or as homemakers (9.6% vs. 1.2%). With regards to the reasons for foreclosure, men more often reported being in this situation due to loss of employment (68.3% vs. 51.2%) and women due to domestic problems (divorce, a death in the family, or other family situations) (29.8% vs. 15.9%).

A variable percentage of participants reported worsening for all the indicators analyzed. The most frequent changes reported were an increase in episodes of anxiety, depression and/or stress (85.7%), and worse sleep patterns (82.6%). Also, 42.8% reported a deterioration of a pre-existing chronic illness, and 40.8% increased their use of medication (Table 2). 

A higher percentage of men than of women stated that their consumption of alcohol had increased (21% vs. 6.2%; *p* = 0.002) and that they did less physical exercise (24.4% vs. 14.2%; *p* = 0.065). For their part, women presented a higher percentage of worsening chronic conditions (53.6% vs. 26.7%), higher consumption of medication, both psychotropic (38.1% vs. 23.1%) and other (47.9% vs. 29.9%), and higher use of health services, both primary care and emergency rooms or mental health services, with their use of the latter being three-fold that of men (24.6% vs. 8.9%; *p* = 0.005). 

Table 3 presents the association between socio-demographic, socio-economic and process variables for each of the health indicators of the total sample, and differentiated for men and women. It only includes the variables that reached significance in the previous step of the analysis. 

Table 4 contains the results of the multivariate analysis for each health indicator by socio-demographic, socio-economic, and process variables. Marital status is related to several health indicators. People who were not married proportionally suffered from a chronic disease much less than those who were married (OR = 0.39, 95% CI: 0.17–0.88, OR = 0.29, 95% CI: 0.12–0.65). This also occurred for the consumption of medication, being lower for singles (OR = 0.76, 95% CI: 0.37–1.55), and for respondents who were divorced, separated or widowed (OR = 0.51, 95% CI: 0.23–1.07). However, married persons had a lower likelihood of having increased smoking, compared to unmarried people (OR = 1.63, 95% CI: 0.77–3.42, OR = 2.77, 95% CI: 1.29–6.04).

The stage of the process is another important variable contributing to our results. People who were already in the legal stage of the process were more than twice as likely to visit an emergency room (OR = 2.36, 95% CI: 1.16–4.90), to smoke more (OR = 2.02, 95% CI: 1.09–3.80), to eat less vegetables (OR = 2.49, 95% CI: 1.29–4.92), and almost twice as likely to consume psychotropic medications (OR = 1.88, 95% CI: 0.98–3.66), as compared to people who were in the initial stages of the process. Another important feature of the process in our results was the main reason for the foreclosure. When the eviction was due to family problems, the probability of having anxiety, depression, or stress (OR = 6.51, 95% CI: 1.58–60.11) was greater than for other causes. The same occurred with regard to alcohol consumption, which increased when the eviction was due to loss of income (OR = 2.02, 95% CI: 0.59–6.32) or domestic problems (OR = 3.17, 95% CI: 1.09–9.31). On the contrary, when the eviction was due to economic problems (OR = 0.34, 95% CI: 0.12–0.82) or domestic problems (OR = 0.82, 95% CI: 0.40–1.72), visits to the primary care services were less frequent, compared to the primary cause, i.e., loss of employment.

Other independent variables were associated with certain health indicators, such as age, education level, or income. Sex was associated with the majority of the health indicators analyzed, being a fundamental variable for explaining the associations between the foreclosure process and health.

## 4. Discussion 

This work adds scientific evidence to the study of the impact of foreclosure processes on people’s health, exploring the socio-demographic, socio-economic and process characteristics that are associated with worsening selected health indicators. Unlike other quantitative studies [26,27,28], which have focused on comparing health indicators between two groups (persons affected by foreclosures and the general population), this work makes an intra-group analysis with an in-depth examination of profiles and their relation to health indicators. 

This investigation confirms and extends the scientific evidence documenting the negative effects of foreclosure processes on the physical and mental health of people who lose their homes [32]. Moreover, this deterioration of health and its determinants seems to have a direct relation to the process of foreclosures reported by the affected persons. Among the health indicators used, the most prevalent effect was an increase in episodes of anxiety, depression and/or stress. Mental health is the focus area of most of the studies [18,19,20,21,33,34,35,36], and all of them highlight its deterioration among mortgage holders or tenants who are behind on their payments [18,27,33]. In particular, an increase in symptoms of depression has been noted [20,22,36]. The second most prevalent indicator of health deterioration in our study was sleep patterns, commonly associated with mental health symptoms [37]. Other studies on foreclosure processes and their effects on health have similarly noted this, such as the study by Novoa et al. [27], carried out in Barcelona, which reported higher rates of sleep deprivation compared with the general population. Moreover, the qualitative study by Ruiz [25] which draws up an ethnography of suffering among persons going through a foreclosure, finds that in addition to disturbed sleep patterns, there are constant mood swings, sadness and irritability, isolation, and blame and social stigma also present, among other manifestations.

The differences of the most prevalent health indicators by sex relate to gender patterns widely recorded in the scientific literature. Thus, among men there is a higher increase in alcohol consumption [38] and greater reduction in physical exercise, while among women there is a higher increase in consumption of medication, including psychotropic medication [39], and in the use of health services [40].

A particular feature of the population going through a foreclosure process is reflected in the socio-economic profile of our sample, the large part of which comprises sectors of the middle class which may be starting to form part of a new profile, the “new poor” emerging as a result of the economic crisis [41,42]. However, our study has found links between disparities in income levels among persons being evicted and some of the health indicators. Thus, the population with lower income levels (500 € per month or less) is more likely to visit the ER more frequently, request mental health appointments, and consume medication—particularly women, for the latter two indicators—than the population with higher income levels (500 € or more). 

The combination of income levels, deprivation, and health are part of a lengthy debate in the public health sector about inequalities, which has emphasized the need for policies aimed at better distribution of wealth [43,44]. Our study found that 52.9% of the sample have seen their income drop from an average of 2000 € before the start of the foreclosure process, to an average of 250 € afterwards. Various studies have found that indebtedness is a risk factor for experiencing some kind of mental disorder, even once the income level has been controlled for [34,45].

In Spain, during our study period, the loss of housing was associated with high unemployment due to the economic crisis. Therefore, during this period, foreclosures have mostly affected the working middle class, and not so much the structurally vulnerable population, as in other countries. This affected working population has suffered, in a very short time, the combination of loss of work due to the crisis, without the possibility of recovery in the short term, together with the situation of foreclosure. These are two potentially catastrophic conditions from a vital and health point of view. Our study shows relevant effects, which affect health to an important degree. These results are compatible with the results of the several studies conducted in Spain on health during foreclosure processes [28,46].

On the other hand, a small number of studies have looked at the impact of the different stages of the process and the length of time experiencing them [18,22,28]. However, the existing studies have pointed out that as the process progresses, the large majority of physical and mental health indicators and their most immediate determinants worsen. Along these lines, the results of this study show that being in the stage of legal proceedings makes for poorer health, as reflected in several indicators, such as increased visits to the emergency room, consumption of psychotropic medication, smoking, and lower consumption of vegetables. Although self-perceived poor health may begin or be accentuated even before the foreclosure process starts, after the first difficulties in meeting repayments are experienced [22], many studies agree that there is further deterioration as time passes and/or the stages of the process progress. In addition, Pevalin [18] has found that the risk of suffering a mental disorder is greater in home-owning situations, which accounts for the great majority of people in the Spanish context and in the study sample. 

Lastly, we point out that among affected persons, domestic problems emerge as the main reason for foreclosure being linked to higher probability of suffering episodes of anxiety, depression and/or stress. Moreover, results showed that marital status was associated with worse outcomes for several health indicators. Whatever the family situation experienced, the fact that persons who perceive domestic problems to be the main reason for foreclosure have poorer health indicators underlines the importance of the family and social support networks, traditionally considered as factors in safeguarding mental and physical health [47]. 

This study presents several methodological considerations and limitations. Firstly, the study focuses on persons belonging to the Stop-Evictions platform. It is the only movement which offers legal advice and mutual support, which is why it is an umbrella movement for the majority of people affected by foreclosure. We do not know whether persons who, while going through a foreclosure process, do not participate in this platform, present different characteristics and could therefore experience other health outcomes. A study in Malaga on the profile of families going through a foreclosure process [48], based primarily on the Stop-Evictions platform for mortgage victims, presents a similar profile to that found in our study and in the one carried out in Catalonia [28]. Nevertheless, given the platform’s assistance role in managing the process, offering legal advice, mutual support and solidarity, consideration must be given to whether persons who do not form part of the platform might exhibit worse health characteristics. Hence, the results of this study may under-represent the real health situation of persons going through a foreclosure process. Secondly, our study has a cross-sectional design. Thirdly, there are several considerations for assessing health with self-reporting questionnaires, even though they are the most common methods, used for their efficiency. Socially desirable, acquiescent, and extreme responses, or distorted self-perceptions, may affect accuracy of information. However, we tried to minimize response bias by using questions already validated in our language and used in national health surveys, and also by piloting the questionnaire. 

## 5. Conclusions

The results of this study highlight worsening health attributable to foreclosure processes in general terms, and a deterioration which varies according to the socio-demographic, socio-economic and process profile of the persons concerned. Our results may contribute to identifying factors that can be used for preventing poor health among populations experiencing foreclosure processes. 

However, further studies will be required to examine this negative relationship in greater depth. Along these lines, it would be helpful to undertake quantitative studies with larger samples, and qualitative studies which examine in greater depth the relationships between adverse structural and material conditions, as well as their effects on the lives and health of the persons concerned, taking into consideration their subjectivities. 

In any case, current housing policies give rise to unsustainable housing situations which can result in prolonging stress and have a negative effect on health. For this reason, measures for putting an end to foreclosure processes are becoming urgently necessary. Similarly, the need to set up health programs to help protect the health of affected persons is also urgent, as has been discussed in detail in another publication of our team [49]. 

## Figures and Tables

**Table 1 ijerph-16-00981-t001:** Socio-demographic, socioeconomic and foreclosure process characteristics of sample by sex.

Characteristics	Total		Men		Women		*p* (χ²)
	*N*	%	*N*	%	*N*	%	
Age							
25–35	58	(28.3)	29	(34.9)	29	(23.8)	0.157
36–50	104	(50.7)	36	(43.4)	68	(55.7)	
51 or older	43	(21.0)	18	(21.7)	25	(20.5)	
Total	205	(100.0)	83	(100.0)	122	(100.0)	
Place of birth							
Spain	183	(91.0)	70	(86.4)	113	(94.2)	0.059
Abroad	18	(9.0)	11	(13.6)	7	(5.8)	
Total	201	(100.0)	81	(100.0)	120	(100.0)	
Marital status							
Married	102	(51.3)	43	(54.4)	59	(49.2)	0.046
Single	52	(26.1)	25	(31.6)	27	(22.5)	
Separated, divorced, widowed	45	(22.6)	11	(13.9)	34	(28.3)	
Total	199	(100.0)	79	(100.0)	120	(100.0)	
Living with partner							
Yes	36	(37.1)	19	(52.8)	17	(27.9)	0.014
No	61	(62.9)	17	(47.2)	44	(72.1)	
Total	97	(100.0)	36	(100.0)	61	(100.0)	
Number of persons in household							
Up to 2	56	(27.6)	23	(28.0)	33	(27.3)	0.402
3 to 5	137	(67.5)	57	(69.5)	80	(66.1)	
More than 5	10	(4.9)	2	(2.4)	8	(6.6)	
Total	203	(100.0)	82	(100.0)	121	(100.0)	
Employment situation							
Employed	51	(26.2)	13	(16.0)	38	(33.3)	0.001
Unemployed	123	(63.1)	61	(75.3)	62	(54.4)	
Retired, Disabled	9	(4.6)	6	(7.4)	3	(2.6)	
Homemaker	12	(6.2)	1	(1.2)	11	(9.6)	
Total	195	(100.0)	81	(100.0)	114	(100.0)	
Socio-professional class							
Management personnel, professional	11	(5.8)	6	(7.5)	5	(4.5)	0.025
Administrative, self-employed. Supervisor	61	(31.9)	33	(41.3)	28	(25.2)	
Skilled manual	61	(31.9)	25	(31.3)	36	(32.4)	
Unskilled manual	58	(30.4)	16	(20.0)	42	(37.8)	
Total	191	(100.0)	80	(100.0)	111	(100.0)	
Level of education							
Up to primary	123	(60.6)	53	(63.9)	70	(58.3)	0.135
Secondary	60	(29.6)	26	(31.3)	34	(28.3)	
University	20	(9.9)	4	(4.8)	16	(13.3)	
Total	203	(100.0)	83	(100.0)	120	(100.0)	
Current income (Euros)							
Up to 500	89	(43.8)	38	(45.8)	51	(42.5)	0.559
501 to 1000	77	(37.9)	28	(33.7)	49	(40.8)	
Over 1000	37	(18.2)	17	(20.5)	20	(16.7)	
Total	203	(100.0)	83	(100.0)	120	(100.0)	
Difficulties getting to the end of the month							
Very difficult	172	(89.1)	69	(85.2)	103	(92.0)	0.312
Quite difficult	17	(8.8)	10	(12.3)	7	(6.3)	
Easy	4	(2.1)	2	(2.5)	2	(1.8)	
Total	193	(100.0)	81	(100.0)	112	(100.0)	
Change in income							
No change or increase	67	(32.7)	22	(26.5)	45	(36.9)	0.170
Drop by 1 level	77	(37.6)	31	(37.3)	46	(37.7)	
Drop by 2 or more levels	61	(29.8)	30	(36.1)	31	(25.4)	
Main reason for foreclosure							
Loss of employment	118	(58.1)	56	(68.3)	62	(51.2)	0.036
Drop in income	36	(17.7)	13	(15.9)	23	(19.0)	
Domestic problems *	49	(24.1)	13	(15.9)	36	(29.8)	
Total	203	(100.0)	82	(100.0)	121	(100.0)	
Stage of process							
Problems paying the mortgage	110	(53.9)	41	(50.0)	69	(56.6)	0.064
Lawsuit (start of legal process)	25	(12.3)	10	(12.2)	15	(12.3)	
Negotiation with bank	25	(12.3)	15	(18.3)	10	(8.2)	
Auction or foreclosure stage	30	(14.7)	14	(17.1)	16	(13.1)	
Other	14	(6.9)	2	(2.4)	12	(9.8)	
Total	204	(100.0)	82	(100.0)	122	(100.0)	
When process began							
Up to 2008	21	(11.7)	8	(10.8)	13	(12.3)	0.118
2009 to 2011	37	(20.6)	10	(13.5)	27	(25.5)	
2012 onwards	122	(67.8)	56	(75.7)	66	(62.3)	
Total	180	(100.0)	74	(100.0)	106	(100.0)	

* Domestic problems refer to couple separation or divorce, death of a family member, and other similar family problems.

**Table 2 ijerph-16-00981-t002:** General perception of changes in health, use of health services, consumption of medication and health-related habits, attributable to foreclosure, by sex.

Since the Beginning of the Foreclosure Process, Have You Noticed Significant Changes Attributable to It?	Increase/Worsen		Total	*p* (χ²)
	*N*	%	*N*	
Anxiety, depression, stress				
Men	69	84.1	82	0.599
Women	105	86.8	121	
Total	174	85.7	203	
Chronic illnesses				
Men	20	26.7	75	0.000
Women	60	53.6	112	
Total	80	42.8	187	
Sleep problems				
Men	65	81.3	80	0.684
Women	101	83.5	121	
Total	166	82.6	201	
Primary healthcare visits				
Men	19	23.8	80	0.077
Women	43	35.5	121	
Total	62	30.8	201	
Mental health appointments				
Men	7	8.9	79	0.005
Women	29	24.6	118	
Total	36	18.3	197	
Visits to Emergency Room				
Men	12	14.8	81	0.062
Women	31	25.8	120	
Total	43	21.4	201	
Consumption of psychotropic med				
Men	18	23.1	78	0.029
Women	43	38.1	113	
Total	61	31.9	191	
Consumption of other med				
Men	23	29.9	77	0.012
Women	57	47.9	119	
Total	80	40.8	196	
Smoking				
Men	32	40.0	80	0.600
Women	41	36.3	113	
Total	73	37.8	193	
Consumption of alcohol				
Men	17	21.0	81	0.002
Women	7	6.2	113	
Total	24	12.4	194	
Consumption of other substances				
Men	6	7.6	79	0.114
Women	3	2.7	112	
Total	9	4.7	191	
Physical exercise and sport (reduced)				
Men	20	24.4	82	0.065
Women	17	14.2	120	
Total	37	18.3	202	
Consumption of vegetables (reduced)				
Men	17	21.0	81	0.295
Women	33	27.5	120	
Total	50	24.9	201	

**Table 3 ijerph-16-00981-t003:** Risk factors associated with self-perceived changes in health (Univariate Regression Models).

Changes in Health	Total (*N* = 205)			Men (*N* = 83)			Women (*N* = 122)		
	*p*-Value	OR	95%CI	*p*-Value	OR	95% CI	*p*-Value	OR	95% CI
Anxiety, Depression, Stress									
Place of birth									
Abroad		1			1			1	
Spain	0.02	3.64	(1.22, 10.86)	0.75	1.31	(0.25, 6.98)	0.00	11.22	(2.24, 56.26)
Number of persons in household									
Up to 2		1			1			1	
More than 2	0.19	1.74	(0.76, 3.96)	0.03	3.79	(1.11, 12.92)	0.83	0.87	(0.26, 2.93)
Main reason for foreclosure									
Loss of employment	0.03	1		0.91	1		0.99	1	
Drop in income	0.22	0.57	(0.23, 1.41)	0.67	0.73	(0.17, 3.12)	0.89	0.91	(0.26, 3.26)
Domestic problems	0.04	9.06	(1.17, 70.16)		NA		0.95	0.96	(0.32, 2.91)
Chronic Illnesses									
Age									
25–35	0.03	1		0.86	1		0.02	1	
36–50	0.16	0.60	(0.29, 1.23)	0.63	0.75	(0.23, 2.44)	0.21	0.56	(0.23, 1.38)
51 or older	0.23	1.71	(0.72, 4.10)	0.99	1.01	(0.26, 3.96)	0.12	2.70	(0.78, 9.35)
Marital status									
Married	0.01	1		0.26	1		0.02	1	
Single	0.03	0.43	(0.20, 0.93)	0.35	0.57	(0.17, 1.88)	0.04	0.36	(0.13, 0.96)
Separated, divorced, widowed	0.00	0.31	(0.14, 0.69)	0.14	0.19	(0.02, 1.67)	0.01	0.33	(0.13, 0.80)
Main reason for foreclosure									
Loss of employment	0.67	1		0.04	1			1	
Drop in income	0.72	1.16	(0.52, 2.61)	0.21	0.26	(0.03, 2.19)	0.22	1.93	(0.68, 5.49)
Domestic problems	0.38	1.39	(0.67, 2.85)	0.05	3.60	(1.01, 12.81)	0.94	0.97	(0.41, 2.26)
Sleep Problems									
Level of education									
Secondary or above		1			1			1	
Primary or lower	0.29	0.66	(0.30, 1.43)	0.36	1.71	(0.55, 5.33)	0.04	0.29	(0.09, 0.94)
Number of persons in household									
Up to 2		1			1			1	
More than 2	0.01	2.63	(1.23, 5.58)	0.03	3.79	(1.11, 12.92)	0.06	2.63	(0.97, 7.09)

NA: Not applicable due to the low number of cases.

**Table 4 ijerph-16-00981-t004:** Multivariate penalized logistic regression models for changes in health, use of health services, consumption of medication, and health related habits.

Changes in Health		*p*-Value	OR	95% CI
Anxiety, Depression, Stress				
Main reason for foreclosure	Loss of employment		1	
	Drop of income	0.265	0.60	(0.25, 1.50)
	Domestic problems	0.006	6.51	(1.58, 60.11)
Chronic Illnesses				
Married status	Married		1	
	Single	0.023	0.39	(0.17, 0.88)
	Separated, divorced, widowed	0.002	0.29	(0.12, 0.65)
Sex	Men		1	
	Women	0.001	3.21	(1.61, 6.61)
Level of education	Primary or lower		1	
	Secondary or above	0.058	1.88	(0.98, 3.68)
Employment situation	Employed		1	
	Unemployed	0.024	0.42	(0.19, 0.89)
	Non-active	0.325	0.56	(0.17, 1.78)
Sleep Problems				
Number of persons in household	Up to 2		1	
	More than 2	0.012	2.62	(1.24, 5.54)
Primary Healthcare Visits				
Age	25–34		1	
	35–50	0.270	1.52	(0.73, 3.31)
	51 or older	0.048	2.41	(1.01, 5.90)
Main reason for foreclosure	Loss of employment		1	
	Drop in income	0.016	0.34	(0.12, 0.82)
	Domestic problems	0.632	0.84	(0.40, 1.72)
Sex	Men		1	
	Women	0.068	1.82	(0.96, 3.57)
Mental Health Appointments				
Sex	Men		1	
	Women	0.003	3.40	(1.49, 8.69)
Current income (euros)	Up to 500			
	501 to 1000	0.038	0.43	(0.18, 0.95)
	More than 1000	0.098	0.41	(0.12, 1.17)
Visits to Emergency (ER)				
Stage of process	Problems paying mortgage		1	
	Legal process	0.018	2.36	(1.16, 4.90)
Sex	Men		1	
	Women	0.044	2.13	(1.02, 4.68)
Consumption of Psychotropic Medication				
Stage of process	Problems paying mortgage		1	
	Legal process	0.056	1.88	(0.98, 3.66)
Sex	Men		1	
	Women	0.012	2.34	(1.20, 4.72)
Consumption of Other Medication				
Married status	Married		1	
	Single	0.453	0.76	(0.37, 1.55)
	Separated, divorced, widowed	0.076	0.51	(0.23, 1.07)
Sex	Men		1	
	Women	0.006	2.38	(1.28, 4.54)
Smoking				
Stage of process	Problems paying mortgage		1	
	Legal process	0.026	2.02	(1.09, 3.80)
Married status	Married		1	
	Single	0.199	1.63	(0.773, 3.419)
	Separated, divorced, widowed	0.009	2.77	(1.288, 6.041)
Consumption of Alcohol				
Main reason for foreclosure	Loss of employment		1	
	Drop in income	0.248	2.02	(0.59, 6.32)
	Domestic problems	0.034	3.17	(1.09, 9.31)
Level of education	Primary or lower		1	
	Secondary or above	0.057	2.38	(0.98, 6.01)
Sex	Men		1	
	Women	0.001	0.19	(0.07, 0.48)
Consumption of Other Substances				
Age	25–34		1	
	35–50	0.239	0.46	(0.12, 1.69)
	51 or older	0.059	0.12	(0.01, 1.07)
Physical Exercise and Sport				
Married status	Married		1	
	Single	0.193	0.63	(0.31, 1.26)
	Separated, divorced, widowed	0.017	0.40	(0.18, 0.85)
Level of education	Primary or lower		1	
	Secondary or above	0.036	1.90	(1.04, 3.51)
Current income (euros)	Up to 500		1	
	501 to 1000	0.574	1.20	(0.64, 2.28)
	More than 1000	0.095	0.48	(0.19, 1.14)
Consumption of Vegetables				
Stage of process	Problems paying mortgage		1	
	Legal process	0.007	2.49	(1.29, 4.92)

Independent variables included in the model: Age; Sex; Marital Status; Level of education; Employment situation; Current income; Main Reason for foreclosure; Stage of process.

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
