# Peer review of "Associations between Home Foreclosure and Health Outcomes in a Spanish City"

_ijerph, 2019, doi:10.3390/ijerph16060981_

Reviewer 1 Report

I think this paper is interesting, is well structured, has got a clear methodology and contributes reasonable conclusions

Although research references on health and eviction are cited, I miss other ones about evictions in general terms, causes and diversity of situations. In the introduction, there is just a paragraph about this issue. I think you should include references as following. Sincerely, this would allow to include very interesting nuances related the purpose of this paper.

In Méndez, R.; Plaza, J. Crisis inmobiliaria y desahucios hipotecarios in España: Una perspectiva geográfica. BAGE 2016, 71, 99–127, you have a very interesting general outlook from judicial sources and aggregated data.

In Parreño-Castellano, J.M.; Domínguez-Mujica, J.; Armengol-Martín, M.; Pérez García, T.; Boldú Hernández, J. Foreclosures and Evictions in Las Palmas de Gran Canaria during the Economic Crisis and Post-Crisis Period in Spain. Urban Sci. 2018, 2, 109, you can appreciate the diversity of situations from judicial sources with microscale analysis (proceeding by proceeding).

In Gutiérrez, A.; Arauzo-Carod, J.M. Spatial Analysis of Clustering of Foreclosures in the Poorest-Quality Housing Urban Areas: Evidence from Catalan Cities. ISPRS Int. J. Geo-Inf. 2018, 7, 23, the relations between housing vulnerability and foreclosures is stablished from alternative (non-official), secondary data sources.

The paper has got abundant statistical input, with several tables of information. But, the description and analysis of results is brief, in special in relation to the results of logistic regression. I advise a richer presentation of results before starting Discussion. Some paragraphs should be added to get a more interesting interpretation.

Misprints:

-          Page 2, Line 33. Square bracket is omitted in reference 22.

-          Page 2, Line 18. UU?

-          Table 1: Postponement (negotiation with

-          Page 14, Line 47. 41 in not included in References.

Author Response

REVIEWER 1

I think this paper is interesting, is well structured, has got a clear methodology and contributes reasonable conclusions

Although research references on health and eviction are cited, I miss other ones about evictions in general terms, causes and diversity of situations. In the introduction, there is just a paragraph about this issue. I think you should include references as following. Sincerely, this would allow to include very interesting nuances related the purpose of this paper.

In Méndez, R.; Plaza, J. Crisis inmobiliaria y desahucios hipotecarios in España: Una perspectiva geográfica. BAGE 2016, 71, 99–127, you have a very interesting general outlook from judicial sources and aggregated data.

In Parreño-Castellano, J.M.; Domínguez-Mujica, J.; Armengol-Martín, M.; Pérez García, T.; Boldú Hernández, J. Foreclosures and Evictions in Las Palmas de Gran Canaria during the Economic Crisis and Post-Crisis Period in Spain. Urban Sci. 2018, 2, 109, you can appreciate the diversity of situations from judicial sources with microscale analysis (proceeding by proceeding).

In Gutiérrez, A.; Arauzo-Carod, J.M. Spatial Analysis of Clustering of Foreclosures in the Poorest-Quality Housing Urban Areas: Evidence from Catalan Cities. ISPRS Int. J. Geo-Inf. 2018, 7, 23, the relations between housing vulnerability and foreclosures is stablished from alternative (non-official), secondary data sources.

The paper has got abundant statistical input, with several tables of information. But, the description and analysis of results is brief, in special in relation to the results of logistic regression. I advise a richer presentation of results before starting Discussion. Some paragraphs should be added to get a more interesting interpretation.

·         Added a paragraph in the introduction: This impact was not homogeneous, but affected much more the Spanish Mediterranean provinces and regions, such as Andalusia. This greater impact in the Mediterranean regions of the country was due to the enormous role that housing construction has had in the economy and employment in these regions, closely associated with tourist activities. (In Méndez, R., Plaza, J. Real Estate Crisis and Mortgage Evictions in Spain: A Geographical Perspective, BAGE 2016, 71, 99-127).

·         Added a paragraph in the introduction:   For example, in 2007 in Andalucia, which represents 18% of the Spanish population, some 125,000 homes were completed; in comparison, the median for all the Spanish regions was about 25,000 homes finished to build (ATLAS-GACETA).

·         Added a paragraph in the introduction: Likewise, there is evidence indicating that the evictions have not been distributed evenly among the neighborhoods of the Spanish cities, but rather that they have been concentrated mainly in the neighborhoods with the greatest deprivation. And in addition, they have affected the smallest and cheapest houses more frequently. All this leads to consider that low-income families have been the most affected by the crisis of the evictions. Gutiérrez, A.; Arauzo-Carod, J.M. Spatial Analysis of Clustering of Foreclosures in the Poorest-Quality Housing Urban Areas: Evidence from Catalan Cities. ISPRS Int. J. Geo-Inf. 2018, 7, 23)

Misprints:

-          Page 2, Line 33. Square bracket is omitted in reference 22.

·         Done

-          Page 2, Line 18. UU?

·         Done

-          Table 1: Postponement (negotiation with banks)

·         Changed to: Negotiation with Banks.

-          Page 14, Line 47. 41 in not included in References.

·         Corrected

Reviewer 2 Report

This work contributes to the field of displacement and health outcomes by examining the relationship between Spanish eviction processes and resident mental health outcomes. The juxtapositions between men and women, and married and unmarried, add a new and interesting approach to the topic. Additionally, the authors designed their study to examine how different stages of the eviction process correlate with particular health outcomes. 

I recommended that the authors spend more time in the introduction defining what they mean by eviction processes. As written, the authors reference housing as a "speculation asset" (page 2, Line 4) "defaulting on mortgage payments," (page 2, Line 12), and include a note about homes available for rent (page 2, line 6) in Spain. It is not clear from this discussion what is considered to be an eviction. Are the authors looking at displacement following default on mortgage payments? Are the authors including displacement after a tenant fails to pay their rent? This is particularly important for an international audience. While it may be the case that a Spanish audience is familiar with the terminology, an American audience, for example, has a very specific understanding of eviction: it pertains to tenants (individuals with only a leasehold interest in a property). In contrast, an American audience would use the word "foreclosure" to describe the process in which a mortgagee (typically a lender/bank) takes possession of a property following the mortgagor's (typically the individual homeowner/family) failure to pay their monthly mortgage payment. To address this, the authors should make clear the type of displacement they are studying. This will help the authors convey the significance of their study.   

Relatedly, the authors should define/explain the eviction process, giving an overview of the stages in the introduction. The authors state that “the eviction process is complex and usually takes several years” (page 2, line 10) but it is not clear what happens, what makes it complex, or why it takes so long. In light of the fact that the authors draw conclusions between different “stages” of the eviction process and particular health outcomes, it is imperative that the authors define and delineate each stage. The article references “the stage of legal proceedings” (page 13, line 49) and a stage “before the eviction process starts, when the first difficulties in meeting repayments are experienced” (page 14, lines 1 – 2) but it is not clear what is contained in each stage. Describing and delineating the stages will strengthen the conclusion’s assertion that there is a “urgent need” to set up services to help those negatively affected by eviction.

Author Response

Dear reviewer:

·         We have revised and  changed the terminology in the text. Following your suggestions we have adopted “foreclosures” instead of “evictions” in our paper.

·         Also, we have changed the title  to “Foreclosures Processes and Their Influence on Changes in Health according to Socio-Demographic, Socio-Economic and Process Characteristics in a Spanish Sample”.

          We have added a paragraph in the introduction to explain more in detail the process of foreclosures in SpainThe process of foreclosure in Spain begins because home owners have difficulties to pay the mortgage. This is the first phase of the process. During this phase the owners have received letters or telephone communications from the bank claiming the amounts due, and in many cases, threatening eviction. After a variable time, between three and six months according to some sources, if the mortgage payments are not made, then the bank will file a lawsuit in the courts claiming the amounts. Then what we call the "legal" phase of the foreclosure begins. In this phase, a negotiation is established between the bank and home owners. If no solutions are found, the lawsuit prospers, and the court issues a legal ruling, and finally the loss of ownership and eviction from the house. This eviction due to foreclosure may be delayed for various reasons. In many cases, according to Spanish legislation, the obligation to pay the debt is maintained, which due to the costs associated with the non-payment, is much greater than the mortgage.

Reviewer 3 Report

Dear Editor;

-          One can definitely see that there is a considerable amount of work invested in the study and serious intentions behind it. I consider that the descriptive information that is presented in the manuscript may be of value to the literature on eviction. However, the manuscript as such suffers from several major shortcomings that make it difficult to judge the quality of the results and the conclusions.  Maybe a more descriptive approach (cf., von Otter C. et al.  (017 Dynamics of evictions: results from a Swedish database. European Journal of Homelessness, 11, 1–23), could be a way forward (?).

Specific comments to Author

-          Please develop the description of the eviction process in Spain – maybe in a separate background section?

-          Given that you set out to “analyze and describe changes in health indicators of affected

persons, due to the eviction process, and the socio-demographic, socio-economic and process characteristics associated with these changes” (p.2 line:35-37) I suggest you briefly discuss (in the introduction section) the work of for example: Kahlmeter et al (2018) Housing Evictions and Economic Hardship. A Prospective Study. European Sociological Review, Volume 34, Issue 1, 1 February, Pages 106–119;   

-           Desmond M, Kimbro RT (2015) Eviction’s fallout: housing, hardship, and health Social Forces, Volume 94, Issue 1, 1 September 2015, Pages 295–324

-          Please develop the recruitment process behind you sample. Is there any randomness in the procedure? How big is the none-response rate? There seem to be an important selection problem here that need to be discussed (what is actually being estimated?). Your multivariate analysis does not solve this issue.  This is of particular importance given that your study design does not include a comparison group.

-          It is not clear to me what you mean with “participants were asked if they “noticed significant changes attributable to the eviction process" (p. 3). I suggest you include the question and answer alternative in the manuscript.

-          Given that you consistently report problems of having small N in your analysis (e.g., table 3-4; NA: not applicable due to low number of cases) you need to use methods that have been specially developed for this kind of situations cf:  King  &  Zeng,  2001, King, G., & Zeng, L. (2001). Logistic regression in rare events data. Political analysis, 9(2), 137-163;  penalized maximum likelihood  estimations (Devika, S., Jeyaseelan, L., & Sebastian, G. (2016). Analysis of sparse data in logistic regression in medical research: A newer approach. Journal of postgraduate medicine, 62(1), 26.

-           Multivariate analysis need to be theory driven, besides the problems of not using a proper analytical tool with the issue of rare events being endemic to the data, you may be having problems of overfitting your model (Peduzzi, Peter; Concato, John; Kemper, Elizabeth; Holford, Theodore R.; Feinstein, Alvan R. (1996). "A simulation study of the number of events per variable in logistic regression analysis". Journal of Clinical Epidemiology. 49 (12): 1373–1379).

-          You manuscript would be strengthen if you are able to discuss some alternative interpretations of you results. When it comes to causality issues (p.14 line 25) a suggestion might be: Stenberg S.-Å.(1991). Evictions in the welfare state: an unintended consequence of the Swedish policy?

Specific comments to Author

-          Please develop the description of the eviction process in Spain – maybe in a separate background section?

-          Given that you set out to “analyze and describe changes in health indicators of affected

persons, due to the eviction process, and the socio-demographic, socio-economic and process characteristics associated with these changes” (p.2 line:35-37) I suggest you briefly discuss (in the introduction section) the work of for example: Kahlmeter et al (2018) Housing Evictions and Economic Hardship. A Prospective Study. European Sociological Review, Volume 34, Issue 1, 1 February, Pages 106–119;   

-           Desmond M, Kimbro RT (2015) Eviction’s fallout: housing, hardship, and health Social Forces, Volume 94, Issue 1, 1 September 2015, Pages 295–324

-          Please develop the recruitment process behind you sample. Is there any randomness in the procedure? How big is the none-response rate? There seem to be an important selection problem here that need to be discussed (what is actually being estimated?). Your multivariate analysis does not solve this issue.  This is of particular importance given that your study design does not include a comparison group.

-          It is not clear to me what you mean with “participants were asked if they “noticed significant changes attributable to the eviction process" (p. 3). I suggest you include the question and answer alternative in the manuscript.

-          Given that you consistently report problems of having small N in your analysis (e.g., table 3-4; NA: not applicable due to low number of cases) you need to use methods that have been specially developed for this kind of situations cf:  King  &  Zeng,  2001, King, G., & Zeng, L. (2001). Logistic regression in rare events data. Political analysis, 9(2), 137-163;  penalized maximum likelihood  estimations (Devika, S., Jeyaseelan, L., & Sebastian, G. (2016). Analysis of sparse data in logistic regression in medical research: A newer approach. Journal of postgraduate medicine, 62(1), 26.

-           Multivariate analysis need to be theory driven, besides the problems of not using a proper analytical tool with the issue of rare events being endemic to the data, you may be having problems of overfitting your model (Peduzzi, Peter; Concato, John; Kemper, Elizabeth; Holford, Theodore R.; Feinstein, Alvan R. (1996). "A simulation study of the number of events per variable in logistic regression analysis". Journal of Clinical Epidemiology. 49 (12): 1373–1379).

-          You manuscript would be strengthen if you are able to discuss some alternative interpretations of you results. When it comes to causality issues (p.14 line 25) a suggestion might be: Stenberg S.-Å.(1991). Evictions in the welfare state: an unintended consequence of the Swedish policy?

Author Response

Dear reviewer:

·         We have added a paragraph in the introduction to explain more in detail the process of foreclosures in Spain: The process of foreclosure in Spain begins because home owners have difficulties to pay the mortgage. This is the first phase of the process. During this phase the owners have received letters or telephone communications from the bank claiming the amounts due, and in many cases, threatening eviction. After a variable time, between three and six months according to some sources, if the mortgage payments are not made, then the bank will file a lawsuit in the courts claiming the amounts. Then what we call the "legal" phase of the foreclosure begins. In this phase, a negotiation is established between the bank and home owners. If no solutions are found, the lawsuit prospers, and the court issues a legal ruling, and finally the loss of ownership and eviction from the house. This eviction due to foreclosure may be delayed for various reasons. In many cases, according to Spanish legislation, the obligation to pay the debt is maintained, which due to the costs associated with the non-payment, is much greater than the mortgage.

  We have introduced a paragraph based on the works of  Kahlmeter and Desmond 

  As for the recruitmente process:

Participants were recruited among people belonging to the platform "Stop-Desahucios" (Stop -Evictions) of Granada and attending their weekly meetings. "Stop-Desahucios", is the Spanish platform of support for people in a process of foreclosure or eviction. This platform is organized in "assemblies". These assemblies are attended by people in the process of eviction to obtain legal support, advice and support in the negotiation with the bank. In the city of Granada there are seven permanent assemblies for the nine neighborhoods of the city; two neighborhoods are very small, and therefore are linked to other neighborhoods and are represented by the same assembly. As a rule, these assemblies meet once a week, on a fixed day and hour of the week for each assembly. In the 13 months that our field work lasted, there were active some 250 foreclosures in the courts of Granada. Therefore, the number of "foreclosures" included in our study is large, compared to the "eviction population of the city".

 For the recruitment we followed loosely (we could not do it strictly) the methodology called "Respondent-driven sampling" (Sordo L1, Pérez-Vicente S, Rodríguez Del Águila MM, Bravo MJ, Respondent-driven sampling for the study of difficult access populations, Lisa G. Johnston and Keith Sabin. Sampling hard-to-reach populations with respondent driven sampling -to-reach populations with respondent driven sampling Methodological Innovations Online (2010) 5 (2) 38-48). This is a set of proposed methods for sampling populations that are difficult to access. The recruitment began in the moments before the weekly meetings of the assemblies, presenting the study before the attendees and inviting them to participate. From there, subjects were recruited during several weeks. The sampling ended when the size of the sample approached the target number initially estimated, and the number of subjects recruited was so small that it indicated the end of the subjects available to participate. 

The subjects are representative of the population of people in eviction of Granada. We do not know if people who, despite going through this process, do not attend these assemblies, differ in their characteristics from those that do, and therefore could present other health outcomes. A study carried out in Malaga (a city very close to Granada) on the profile of families in eviction, mainly from the Platform of Affected by Mortgages, and organizations such as the Office of Mortgage Intermediation, Community Social Services, Diocesan Cáritas and Red Cross, presents a profile very similar to the one found in our study (Arredondo R, Palma MO) Approach to the reality of evictions Profile and characteristics of families in eviction process in the city of Málaga. Alternativas. Social Work 2013; 20: 113-40); and the same in the one carried out in Catalonia (Observatory DESC and PAH Impacts of the mortgage crisis in health care: In Valladolid, coordinator Housing emergency in Catalonia Impact of the hypochondrial crisis in the health care els drets dels infants Barcelona: Observatori DESCi PAH, 2015. pp. 96-103.22).

On the other hand, given the role of mutual support and solidarity exercised by the platform "Stop-Desahucios", we believe that those people who (losing their usual home) are not part of it could have worse health, so our results could be be underrepresenting the real state of health of people in the process of eviction.

·         We have included the specific question in the methods section: Specifically they were asked: Have you observed, since the beginning of the process, significant changes attributable to the process of foreclosure?

 ·         With respect to statistical analysis, we were aware that we have a small sample size, although not so small. To do the multivariate analysis, and avoid problems of sparse data and possible overfitting, we used a very cautious procedure. We entered in the multivariate logistic regression models, for each dependent variable, only those variables that showed significance or close to statistical significance in the univariate logistic regression. This has limited the number of independent variables in the multivariate models, to no more than three independent variables. In addition, we checked the distribution of the sample between the two categories of each dependent variable, to verify that it was not an extreme distribution. Finally, we reduce the levels of independent variables, as much as possible without losing their explanatory capacity. Therefore, we think that our analysis was sufficiently robust.

·         However, we found a good idea your suggestion to repeat the analysis using a special statistical technique to prevent sparse data problems. We have repeated the analysis using penalized logistic regression. We used the Firth regression procedure in SPSS, through an extension module of R. As can be seen in the table of results, most of the models remain very similar, modestly varying the magnitude of the Odds ratios and 95CIs. 

·         We have made the analysis based on our hypotheses regarding the impact of foreclosures on health. But the literature is so scarce, that we do not have strong criteria to build the models in a totally specific way.

·         We have modified a paragraph in the discussion section, to discuss other interpretations, as suggested..: